# The Application of Biomedicine in Chemodynamic Therapy: From Material Design to Improved Strategies

**DOI:** 10.3390/bioengineering10080925

**Published:** 2023-08-03

**Authors:** Bingwei Cheng, Dong Li, Changhong Li, Ziqi Zhuang, Peiyu Wang, Gang Liu

**Affiliations:** 1State Key Laboratory of Vaccines for Infectious Diseases, Xiang An Biomedicine Laboratory, National Innovation Platform for Industry-Education Integration in Vaccine Research, State Key Laboratory of Molecular Vaccinology and Molecular Diagnostics, Center for Molecular Imaging and Translational Medicine, School of Public Health, Xiamen University, Xiamen 361102, China; chengbingwei2021@163.com (B.C.); lichanghong960223@163.com (C.L.); 32620221150925@stu.xmu.edu.cn (Z.Z.); gangliu.cmitm@xmu.edu.cn (G.L.); 2College of Ocean Food and Biological Engineering, Jimei University, Xiamen 361021, China

**Keywords:** chemodynamic therapy, Fenton/Fenton-like reaction, nanomedicine materials, tumor microenvironment

## Abstract

Chemodynamic therapy (CDT) has garnered significant interest as an innovative approach for cancer treatment, owing to its notable tumor specificity and selectivity, minimal systemic toxicity and side effects, and absence of the requirement for field stimulation during treatment. This treatment utilizes nanocatalytic medicines containing transitional metals to release metal ions within tumor cells, subsequently initiating Fenton and Fenton-like reactions. These reactions convert hydrogen peroxide (H_2_O_2_) into hydroxyl radical (•OH) specifically within the acidic tumor microenvironment (TME), thereby inducing apoptosis in tumor cells. However, insufficient endogenous H_2_O_2_, the overexpressed reducing substances in the TME, and the weak acidity of solid tumors limit the performance of CDT and restrict its application in vivo. Therefore, a variety of nanozymes and strategies have been designed and developed in order to potentiate CDT against tumors, including the application of various nanozymes and different strategies to remodel TME for enhanced CDT (e.g., increasing the H_2_O_2_ level in situ, depleting reductive substances, and lowering the pH value). This review presents an overview of the design and development of various nanocatalysts and the corresponding strategies employed to enhance catalytic drug targeting in recent years. Additionally, it delves into the prospects and obstacles that lie ahead for the future advancement of CDT.

## 1. Introduction

Cancer, as a major public health problem, causes approximately 10 million deaths worldwide every year due to its high incidence rate and mortality [1]. However, for patients with early-stage (stage I/II) cancer, traditional cancer treatments such as surgery, chemotherapy, and radiotherapy are less effective and have unavoidable side effects [2,3,4]. Therefore, the development of innovative and effective cancer therapies with improved efficacy and fewer side effects is both urgent and challenging.

Reactive oxygen species (ROS) is a general term for a class of chemically active molecules or ions with high oxidation activity, including hydrogen peroxide (H_2_O_2_), superoxide anion (O_2_^−^), hydroxyl radical (•OH) and singlet oxygen (^1^O_2_), peroxy hydroxyl radical (•OOH), etc., which plays an important role in various physiological processes of biological systems [5]. A large number of studies have shown that the overgenerated ROS could break the redox hemostasis and cause a series of biochemical reactions, such as decreased mitochondrial membrane potential, DNA breakage, cytoskeleton contraction, chromatin condensation, etc., leading to cancer cells death [6,7]. Since cancer cells are more sensitive to ROS levels, ROS generation is considered to be an effective way to kill tumor cells selectively.

Currently, the applications of ROS-based tumor therapies mainly including photodynamic therapy (PDT), chemodynamic therapy (CDT), and sonodynamic therapy (SDT) have attracted significant attention from researchers [8,9]. Among these treatment modalities, various methods can generate excessive ROS. In PDT treatment, the photosensitizer generates ROS under the action of near-infrared irradiation, and the local ROS burst promotes cell apoptosis. The cavitation effect, sonoluminescence, and piezoelectric effect are the main mechanisms of ROS generation in SDT. At the same time, there are other ways to increase ROS levels in tumors. For instance, Wang et al. reported a kind of ultrathin two-dimensional metal-organic framework (Cu-TCPP), which was endowed with selectively producing ^1^O_2_ in the tumor microenvironment for tumor therapy [10]. Firstly, the acidic H_2_O_2_ peroxidized the tetrakis(4-carboxyphenyl) porphyrin (TCPP) ligand, and then it was reduced to peroxyl radicals under the action of peroxidase (POD)-like nanosheets and Cu^2+^. Finally, a spontaneous recombination reaction based on the Russell mechanism occurred to generate ^1^O_2_. In addition, Cu-TCPP also consumed glutathione (GSH) through a cyclic oxidation mechanism. Based on the above two “magic weapons”, Cu-TCPP nanosheets efficiently and selectively destroyed tumors.

CDT, proposed by Bu and co-workers in 2016, has emerged as a fascinating research area in recent years. This is primarily due to its remarkable specificity and selectivity for tumors, minimal systemic toxicity and side effects, and the absence of a requirement for field stimulation during treatment [11,12]. All the facts demonstrated that CDT not only had the potential to enhance the efficacy of cancer treatment but could also be used for bacterial infection treatments [13]. Generally, CDT utilizes peroxidase-like catalysis, metallic catalysis, or Fenton and Fenton-like reactions to significantly enhance intracellular ROS. Herein, transitional metal-based nanocatalytic medicines containing some metal ions (e.g., Fe, Cu, Mn, Co, V, Pd, Ag, Mo, Ru, W, and Ce) are utilized to release metal ions in tumor cells and then trigger Fenton and Fenton-like reactions to convert H_2_O_2_ into •OH with higher toxicity in an acidic tumor microenvironment (TME), thus inducing the death of tumor cells [14]. However, the performance of CDT is limited by some natural factors, which makes it difficult to be widely applied. Firstly, there are insufficient endogenous H_2_O_2_ in tumors with a content of 50–100 µmol L^−1^, which is insufficient to meet the ideal CDT therapeutic efficiency [15]. Secondly, the produced oxidative •OH could be captured and neutralized by the overexpressed reducing substances in the TME, such as GSH and hydrogen sulfide (H_2_S), resulting in an unsatisfactory therapeutic efficiency [16,17]. Thirdly, the weak acidity (pH 6.5–7.0) of solid tumors is not suitable for Fenton and Fenton-like reactions, which are required to be remodeled to provide more suitable reaction conditions [18]. Up to now, a variety of nanozymes and strategies have been designed and developed in order to solve these problems (Figure 1) [12,19]. This review aims to provide an overview of the design and development of different types of nanozymes as well as strategies to improve CDT in recent years. Finally, the opportunities and challenges for the future development of CDT are also discussed to promote clinical translation.

## 2. Fenton/Fenton-like Reagent

Nowadays, the research into efficient nanozymes for Fenton and Fenton-like reactions has attracted extensive attention due to the wide application of CDT in cancer therapy. The choice of nanomaterials (iron-based materials, copper-based materials, and other metal-based materials) is crucial because of the irreplaceable role of catalysts in Fenton and Fenton-like reactions.

### 2.1. Iron-Based Fenton Reagent

In recent years, many nanozymes containing catalytically active ions have been designed based on the Fenton reaction and the Fenton-like reaction. So far, iron-based materials have been widely used in the biological field and have shown excellent biosafety and biocompatibility [20,21]. Various iron-based nanomaterials have been extensively studied to enhance the efficacy of CDT, such as ferroferric oxide (Fe_3_O_4_) [22], ferrous disulfide (FeS_2_) [23], ferric oxide (Fe_2_O_3_) [24], Fe-metal organic frameworks (Fe-MOFs) [25], and Fe-doped nanoagents [26]. A Fenton reaction is often catalyzed by Fe^2+^ or Fe^3+^, which decomposes H_2_O_2_ to produce radical •OH, resulting in the destruction of lipids, proteins, and DNA. The reactions (Equations (1) and (2)) are shown below:Fe^2+^ + H_2_O_2_ → Fe^3+^ + •OH + OH^−^(1)
Fe^3+^ + H_2_O_2_ → Fe^2+^ + •OOH + H^+^(2)

As one of the most classical H_2_O_2_-responsive nanozymes, Fe_3_O_4_ nanoparticles become an effective POD-like enzyme with the Fe^2+^/Fe^3+^ redox cycle. As the reaction site of nanozymes, Fe^2+^ on the surface of Fe_3_O_4_ nanoparticles reacted with the overexpressed H_2_O_2_ in TME to produce •OH, which was then converted to Fe^3+^ in the oxidized state, leading to severe tumor damage. In this reaction process, the oxidation property of the generated •OOH was lower than that of the generated •OH, and the rate of the generated •OH was relatively faster [27,28]. For instance, Du et al. successfully constructed kinds of hollow Fe_3_O_4_ mesocrystals to realize the high level of •OH in the tumor region and a low expression of heat shock proteins, resulting in a self-enhanced antitumor efficacy between magnetic hyperthermia and CDT [29]. The therapeutic efficiency of nanomaterials was greatly restricted due to the low iron loading and cumbersome releasing route. Polydopamine (PDA) has been widely studied as a CDT agent owing to its high biocompatibility and radical scavenging strong chelation with Fe^2+^. Recently, Xiao et al. prepared a multifunctional hybrid nanozyme (PDA/Fe_3_O_4_), which was retained in infected sites via an external magnet (Figure 1a) [30]. PDA with redox-active catechol moieties supplied oxygen with electrons to produce H_2_O_2_, which was then decomposed by decorated ultrasmall Fe_3_O_4_ to generate more •OH. Surprisingly, PDA/Fe_3_O_4_ was equipped with simultaneous H_2_O_2_-self-sufficient •OH generation and GSH depletion to improve the CDT efficacy. In another work, Luo et al. synthesized a novel nanoplatform of Fe_3_O_4_@PDA@BSA-Bi_2_S_3_ nanoparticles via the amidation between the carboxyl and amino groups to promote the efficiency of Fenton reaction in cancer cells [31]. In this system, the Fe_3_O_4_ nanoparticles were not only capable of triggering Fenton reactions to generate highly •OH from the innate H_2_O_2_ in the TME but were also employed as the magnetic resonance imaging (MRI) contrast agent for the precise cancer diagnosis. Specifically, PDA promoted the long-term Fenton reactions and tumor apoptosis by preventing oxidation of Fe_3_O_4_, which reacted with intracellular H_2_O_2_ to produce •OH and further suppressed tumor growth. Moreover, computed tomography (CT) contrast was also enhanced due to the significant X-ray attenuation coefficient of Bi_2_S_3_, enabling the simultaneous CDT and PDT treatment of tumors. On the other hand, this also contributed to the regulation of the CDT treatment process, minimizing the biotoxicity and improving diagnostic and therapeutic capabilities. Eventually, the present study demonstrated a novel and viable approach for synthesizing composite theranostic nanoplatforms. 

In addition to Fe_3_O_4_, other iron-based materials are also endowed with superb Fenton reaction catalysis. For instance, Wu et al. developed hollow porous carbon-coated FeS_2_ (HPFeS_2_@C) nanocatalysts for triple-modal imaging-guided synergistic tumor starvation therapy (ST), photothermal therapy (PTT), and CDT [32]. Interestingly, glucose oxidase (GOx) in the multifunctional nanocatalysts reacted with glucose in the TME to increase the H_2_O_2_ concentration, which improved the production of •OH. Meanwhile, tannic acid (TA) was effectively released for reducing Fe^3+^ to Fe^2+^ under near-infrared (NIR) laser exposure, thereby accelerating the Fenton reaction. In addition, the photothermal effect induced by NIR lasers also increased the catalytic efficiency. Furthermore, our group reported a phthalocyanine-iron-based complex FeS_2_@PcD, program-regulated by the dual factors of gluconic acid (H^+^) and H_2_O_2_ in the TME (Figure 1b) [33]. After the efficient internalization into cancer cells, FeS_2_@PcD reacted with excessive H^+^ to generate Fe^2+^. Fe^2+^ then catalyzed intracellular H_2_O_2_ to produce •OH for CDT and simultaneously generate Fe^3+^ for further MRI as well as activating the released building block PcD (Figure 1c). Interestingly, the nitrogen atom on the DPA group of PcD specifically chelated Fe^3+^, thereby recovering fluorescence and sonosensitizing activity. Moreover, in vitro data showed that the fluorescence intensity of FeS2@PcD increased 52.21-fold, resulting from the programmable response, whereas the signal of the H^+^ or H_2_O_2_ univariate factor was almost unchanged (Figure 1d). Moreover, FeS_2_@PcD combined with ultrasound irradiation had a significant inhibitory effect on HepG2 tumor-bearing mice tumor, with an inhibition rate of 87.15% (Figure 1e). In summary, our work achieved precise sonodynamic and chemodynamic therapies, providing a novel strategy for CDT-related clinical research. To date, dihydroartemisinin (DHA) has been regarded as a drug candidate for cancer therapy. This mechanism was based on the breakdown of weak endoperoxide bridges by Fe^2+^ to produce toxic ROS, which resulted in facilitating H_2_O_2_-mediated Fenton reactions to generate abundant ROS for effective CDT [34]. Xu et al. prepared an antibacterial nanoagent Fe_2_O_3_@DHA@MPN (FDM) for dual-augmented NIR and DHA antibacterial CDT [35]. As the core, α-Fe_2_O_3_ mesoporous nanorods provided both efficient DHA transport and an additional source of Fe^2+^ used for enhanced CDT. Furthermore, metal-polyphenol networks (MPN) shells released DHA in response to the microenvironment and led to the release of TA and Fe^3+^ simultaneously due to bacterial infection. Consequently, TA reduced Fe^3+^ to Fe^2+^ under NIR laser exposure, which was favorable for H_2_O_2_-mediated CDT as well as DHA-mediated CDT. 

**Figure 1 bioengineering-10-00925-f001:**
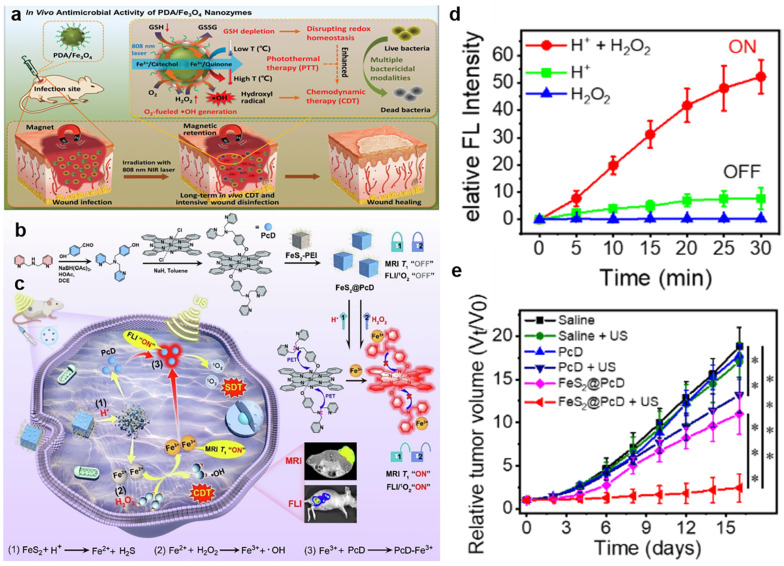
(**a**) The mechanism underlying the effects of PDA/Fe_3_O_4_ hybrid nanozymes as an enhanced CDT agent for bacterial elimination. Reproduced with permission [30]. Copyright 2021, WILEY–VCH. (**b**) Fabrication and programmable mechanism of the FeS_2_@PcD. (**c**) The action process of FeS_2_@PcD in HepG2 cells for sono/chemodynamic therapies. (**d**) Fluorescence intensity of FeS_2_@PcD in the presence of H^+^ and H_2_O_2_. (**e**) Curves showing HepG2 tumor volumes in mice treated with FeS_2_@PcD–mediated SDT. Data were expressed as mean ± SD, n = 5, ** *p* < 0.01, *** *p* < 0.001, **** *p* < 0.0001, analysed by Student’s *t* test. Reproduced with permission [33]. Copyright 2023, Elsevier Ltd.

Recently, MOFs, which are composed of metal ions and organic ligands, have attracted increasing attention in catalysis, energy storage, separation, and cancer therapy [36,37]. Owing to the large pore surfaces, good biodegradability, and biocompatibility, MOFs were widely used as nanoplatforms for drug delivery. Therein, Fe-MOFs (like MIL-101) showed an efficient catalytic effect as a Fenton nanocatalyst. For instance, Chen et al. prepared photosensitizer-integrated nanoagents of MIL-101(Fe)@TCPP for implementing the combined PDT and CDT [38]. In this system, the MIL-101(Fe) MOF transformed endogenous H_2_O_2_ into •OH via Fe ion-catalyzed Fenton reaction, resulting from the intrinsic POD-mimicking activity of MIL-101(Fe)@TCPP. Moreover, the photosensitizer TCPP was covalently connected with MIL-101(Fe) through amide bonds, leading to in situ ^1^O_2_ evolution at the tumor region under light irradiation and amplifying ROS-induced oxidative damage. Similarly, Yin et al. successfully synthesized a cascade biomimetic nanoplatform of CPPO@Fe-porphyrin-MOF@Cancer cell membrane-GOD (C_1_@M@C_2_G) by loading harbors bis(2-carbopentyloxy-3,5,6-trichlorophenyl) oxalate (CPPO) into the Fe-porphyrin-MOF nanoparticle, which was further coated with the cancer cell membrane and grafted GOD on the surface (Figure 2a) [39]. After accumulating in the tumor site, GOD-catalyzed glucose oxidation activated the therapy of starvation, thereby consuming the local glucose and further generating H^+^ and H_2_O_2_, further greatly decreasing the pH of the microenvironment and promoting the massive generation of •OH for efficient CDT. Simultaneously, CPPO, as an energetic reagent, reacted with some other H_2_O_2_ to form high energy states, which excited porphyrin photosensitizers and produced ^1^O_2_ for PDT in situ. Overall, this work combined CPPO-induced PDT, Fenton reaction-based CDT, and GOD-catalyzed ST to synergistically enhance cancer therapy with effective cascade catalysis, and also broaden anticancer pathways.

Additionally, due to properties such as prominent molecular recognition ability and accurate addressability of nucleic acids, DNA-based hydrogels have outstanding advantages in controlling the release of encapsulated cargoes for controlled drug delivery [40,41,42]. Wang et al. prepared a kind of MOF-biomineralized DNA nanosphere (Fe-DNA@ZIF-8) via cascade reactions to implement the intracellular H_2_O_2_ level for enhanced CDT (Figure 2b) [43]. Fe-DNA was composed of Fe^2+^ and DNA molecules by coordination-driven self-assembly, which was then coated by zeolitic imidazolate framework-8 (ZIF-8) for mineralization. At the same time, GOx was encapsulated into ZIF-8 to further form Fe-DNA/GOx@ZIF-8. After being endocytosed by tumor cells, Fe-DNA and GOx were released in the acidic condition, providing Fe^2+^ and Fe^3+^ to activate the Fenton reaction and catalyzing glucose into H_2_O_2_ to enhance CDT effectiveness. In addition, multimetallic alloy nanostructures demonstrated higher catalytic activity than monometallic components owing to the electronic coordination effect, leading to its wide application. For example, Jana et al. fabricated a trimetallic (Pd, Cu, and Fe) alloy nanozyme (PCF-a NEs) with dynamic active-site synergism, which depleted intracellular overexpressed GSH and reduced the cellular self-defense antioxidant mechanism [44]. PCF-a NEs had the synergistic peroxidase-like property to promote •OH in the presence of Cu and Fe, thus boosting the effectiveness of CDT. This work pointed to the potential for using alloy nanozymes for tumor-specific treatments mediated by external stimulus.

To improve the application of iron-based nanoparticles for efficacious CDT, an effective method was to accelerate the conversion efficiency of Fe^2+^ and Fe^3+^. The low indirect band-gap (1.2–1.8 eV) of MoS_2_ made it an effective co-catalyst for the Fenton reaction, which converted Fe(III) into Fe(II). For example, our group synthesized gallic acid (GA)-modified MoS_2_ nanosheets coated with high Fe(III) (MoS_2_@GA-Fe), which were used for photoacoustic (PA) imaging and MRI-guided combined cancer treatment integrating •OH and H_2_S (Figure 2c) [45]. Firstly, GA-Fe(III) reacted with overexpressed GSH to produce GA-Fe(II), which promoted the generation of •OH in the TME. Moreover, after Mo (IV) was oxidized to Mo (VI), S-Mo-S was destroyed, exposing active sites for further reaction cycles and converting unsaturated sulfur atoms to H_2_S with selective cytotoxicity to cancer cells. Additionally, PA imaging for BALB/c nude tumor-bearing mice after intravenous injection of MoS_2_@GA-Fe showed that the signal of the tumor area was enhanced and remained for 24 h, resulting from the structure-based enhanced permeability and retention (EPR) effect (Figure 2d,e). Thus, the TME-responsive “Fenton Nanoreactor” was used as a promising nanomedicine to treat cancer or other diseases and improve patient health. Figure 2(**a**) Schematic representation of the fabrication procedure of C_1_@M@C_2_G NSs and their working mechanism of CL–induced PDT, Fenton reaction–based CDT, and GOD–mediated ST. Reproduced with permission [39]. Copyright 2021, Royal Society of Chemistry. (**b**) Scheme of Fe–DNA/GOx@ZIF–8 for enhanced CDT. Reproduced with permission [43]. Copyright 2021, Elsevier Ltd. (**c**) Theranostic mechanism of the MoS_2_@GA–Fe NPs for PA/MR imaging–guided HCC treatment. (**d**) The PA images and (**e**) intensity of tumor–bearing mice after intravenous injection with MoS_2_@GA-Fe. Reproduced with permission [45]. Copyright 2021, Elsevier Ltd.
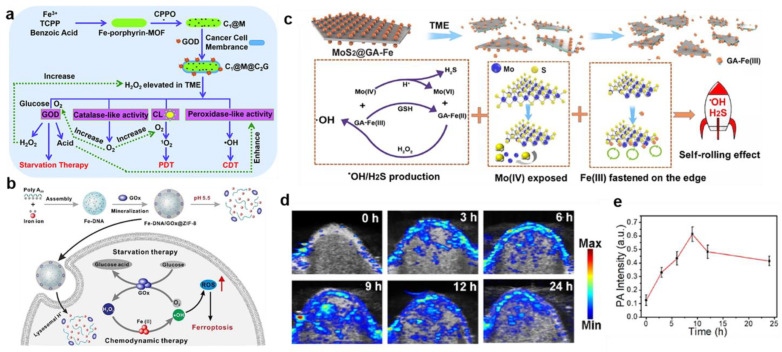


### 2.2. Copper-Based Fenton-like Reagent

In addition to ferrous iron ions, other transition metal ions like Cu^2+^ also have catalytic effects in Fenton and Fenton-like reactions. As a potential alternative to Fe-based nanomaterials, Cu-based nanomaterials have found a variety of applications in many CDT research fields. Cu-catalyzed Fenton-like reactions occur in neutral and very mild acidic conditions, which are less violent than that of Fe-catalyzed Fenton reactions [46]. Moreover, the valence transition between Cu^2+^ and Cu^+^ is usually accompanied by depletion of the excessive GSH in the TME, enhancing the production of •OH to achieve highly efficient CDT [47]. Hence, Cu-based nanomaterials have been extensively employed as Fenton reagents, such as copper sulfide (CuS), copper oxide (CuO), Cu-based MOF, and Cu-doped nanoagents [48,49,50,51]. The Cu-based catalytic reactions (Equations (3) and (4)) are shown below.
Cu^+^ + H_2_O_2_ → Cu^2+^ + •OH + OH^−^(3)
Cu^2+^ + H_2_O_2_ → Cu^+^ + •OOH + H^+^(4)

More and more attention has been paid to CuS nanoparticles because of their low toxicity, their ability to penetrate into biological tissues after being stimulated by NIR light, and their PA and Fenton-like properties [52,53,54]. For instance, Kong and co-workers reported a smart biomimetic enzyme system of CuGP/G, which was composed of the CuS nanoparticle, generation 5 poly (amidoamine) (G5) dendrimer, and GOD for the combination of CDT and PTT (Figure 3a) [55]. Specifically, CuS cores with Fenton-like catalysts were entrapped by positively charged G5, which simultaneously bonded to the negatively charged GOD via electrostatic interactions as a carrier. Additionally, H_2_O_2_ was produced by GOD to generate •OH in the sequential Fenton-like reaction, resulting in efficient tumor treatment. Moreover, the Michaelis–Menten kinetic curve demonstrated that the Michaelis–Menten constant (Km) was 3.75 × 10^−3^ M, and the maximum reaction velocity (Vm) was calculated at 3.6 × 10^−7^ M s^−1^ after incorporating GOD with CuGP (Figure 3b,c), compared to a Fe_3_O_4_-GOD system with the Km of 10.93 × 10^−3^ M and the Vm of 4.22 × 10^−8^ M s^−1^, indicating that the Fenton-like catalytic activity of CuGP/G was better than the Fe_3_O_4_-GOD-based system [56]. Nevertheless, CuS was usually trapped by the endothelial reticular system or rapidly cleared by the kidney, resulting in poor accumulation in tumors. To efficiently transport CuS to tumor cells, Li et al. used macrophages to phagocytose large supramolecular aggregates of CuS for the specific-release nanomedicines in the tumor region (Figure 3d) [57]. The supramolecular aggregates were composed of CuS nanoparticles, which were capped by β-cyclodextrin (β-CD) and ferrocene (Fc), respectively, and self-assembled via β-CD-Fc host–guest interactions. Since macrophages were attracted to tumor tissue with inflammatory properties, nanomedicines were successfully delivered through macrophages and then the β-CD-Fc host–guest pair were dissociated by the oxidation of ferrocene, inducing the disassemble of CuS aggregates into small CuS nanoparticles for further photothermal-enhanced CDT. This work proposed a feasible strategy for specific cancer therapy and new insights into new cell-based medicine carriers.

CuO nanoparticles have aroused great interest in recent years, benefiting from the large surface area, reactive morphology, high catalytic reusability, and superb chemical and thermal stability [58,59]. As mentioned earlier, the rate of •OH production was relatively faster relative to •OOH in Cu-mediated CDT. As a representative CuO-based Fenton-like nanocatalyst in CDT, Jiang et al. reported the synthesis of semiconductor CuO and MoS_2_ nanoflowers to construct MoS_2_-CuO heteronanocomposites through a two-step hydrothermal method (Figure 4a) [60]. MoS_2_-CuO heteronanocomposites were then loaded with bovine serum albumin (BSA) and immunoadjuvant imiquimod (R837) on its surface to further form MoS_2_-CuO@BSA/R837 (MCBR) nanoplatforms, which had an excellent ability to eliminate the primary tumor and release tumor-associated antigens (TAAs). In this system, CuO responded to overexpressed H_2_O_2_ in tumor cells and produced a mass of •OH through Fenton-like reactions. The reactions were further improved by the effect of MoS_2_ due to its high photothermal conversion efficiency. And more importantly, MCBR released TAAs under 808 nm NIR laser irradiation, which combined with R837 to act as an immune-stimulating adjuvant to boost dendritic cells maturation and the release of effector T cells from the lymph node. Consequently, this work achieved an efficient and tumor-specific therapy, providing a meaningful paradigm for future clinical applications. In another study, Xiao et al. prepared an intelligent nanoplatform of CuO_2_@mPDA/DOX-HA (CPPDH) with self-sufficient H_2_O_2_ and GSH depletion for the synergistic chemotherapy, PTT, and CDT (Figure 4b) [61]. The surface of CuO_2_ nanospheres was decorated with mesoporous polydopamine (mPDA) with seed-mediated microemulsion and further loaded with doxorubicin (DOX) via π-π stacking. Moreover, the obtained CuO_2_@mPDA/DOX was constructed by hyaluronic acid (HA) using electrostatic adsorption to finally acquire CPPDH. After entering the intracellular HA environment, CuO_2_ was released to generate Cu^2+^ and H_2_O_2_ triggered by the acidic environment. The Cu^2+^ was then reduced to Cu^+^ by GSH, and subsequently Cu^+^ reacted with intracellular H_2_O_2_ to produce massive •OH for enhanced CDT. Interestingly, CPPDH had excellent targeting ability to tumor cells resulting from the specific binding affinity of HA, whose bonds overexpressed CD44 receptor in the tumor region. Furthermore, the nanoparticles were endowed with a superb effect of PTT due to the strong NIR absorption capacity of PDA, and the light-to-heat conversion efficiency was calculated as 38.39%, which was significantly higher than that of reported photothermal reagents like AuNRs@SiO_2_ (20.8%) [62], MS-BSA (21.8%) [63], and Au-Cu_9_S_5_ nanoparticles (37%) [64].

In addition to the use of CuS and CuO, Cu-MOF also exhibited superb catalytic performance to achieve highly efficient CDT. For example, Tian et al. reported a nanoplatform integrating vitamin K3 into the MOF-199(Cu) to implement a CDT-mediated tumor therapy [65]. The hollow structures derived from MOFs have the virtue of drug loading and delivery with outstanding mass transfer and loading capacity. In a recent study, Cheng et al. proposed a strategy to incorporate Cu^2+^ into the precursors of ZIF-90 to form the Cu^+/2+^ mixed-valence hollow porous structure (Cu/Zn-MOF) through heating treatment, and then Cu/Zn-MOF was integrated with Mn^2+^/MnO_2_ using manganese(II) acetylacetonate(Mn(acac)_2_) to construct a mixed-metal and mixed-valence (Cu^+/2+^/Mn^2+/4+^) structure, which was finally loaded by the photosensitizer indocyanine green (ICG) to prepare ICG@Mn/Cu/Zn-MOF@MnO_2_ (Figure 4c,d) [66]. After being endocytosed by tumor cells, Cu^2+^ and MnO_2_ oxidized GSH to generate a great mass of ROS and Cu^+^ and produced Mn^2+^, which was used to “turn on” MRI. Subsequently, the obtained Cu^+^ and Mn^2+^ catalyzed H_2_O_2_ to produce •OH for effective CDT via the Fenton-like reaction and O_2_ through a synergistic catalytic effect to relieve the hypoxia for the enhanced PDT. Then, the aggregated ICG in the system was not only employed for photothermal imaging (PTI) with excellent photothermal capacity, but also achieved “turn on” fluorescence imaging (FLI) after its release in tumor cells. Irradiated with a single 808 nm NIR, ICG@Mn/Cu/Zn-MOF@MnO_2_ exhibited an obvious temperature increase with an excellent photothermal conversion efficiency of 30.1% and converted the O_2_ generated in situ into ^1^O_2_ for enhancing the effect of PDT. ICG@Mn/Cu/Zn-MOF@MnO_2_ was expected as a versatile platform of PTI/FLI/MRI trimodality imaging due to the high photothermal conversion, ICG release capability, and the generation of Mn^2+^ that responded to TME.
Figure 4(**a**) Schematic illustration of the therapeutic mechanisms of MoS_2_–CuO@BSA/R837 nanoplatforms for synergistic PTT/CDT/immunotherapy. Reproduced with permission [60]. Copyright 2021, Elsevier Ltd. (**b**) Schematic illustration of the synthesis of CPPDH and the schematic diagram of the therapeutic mechanism. Reproduced with permission [61]. Copyright 2021, American Chemical Society. (**c**) Schematic illustration of the synthesis of ICG@Mn/Cu/Zn–MOF@MnO_2_. (**d**) The scheme of therapeutic mechanism for PTI/FLI/MRI–guided ROS–augmented synergistic PTT/PDT/CDT. Reproduced with permission [66]. Copyright 2021, WILEY–VCH.
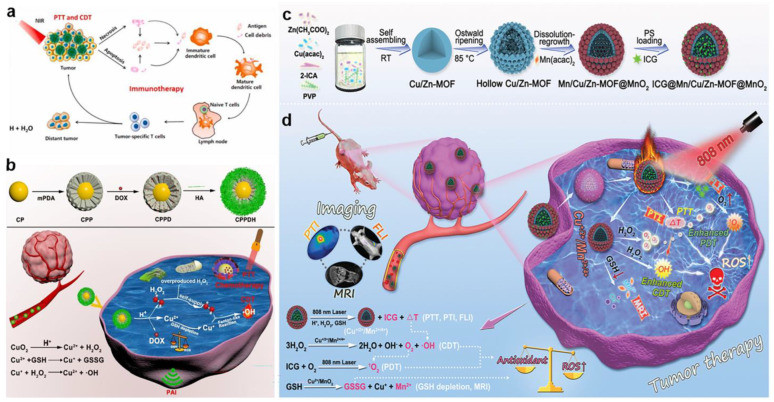


### 2.3. Other Metal-Based Fenton-like Reagents 

In addition to Fe-and Cu-based Fenton reagents, other metals have also been used in Fenton and Fenton-like reactions, such as Mn [67], Ti [68], Ag [69], Pt [70], V [71], and Ru [72]. For Mn-based Fenton reagents, manganese oxide (MnOx) triggers the generation of oxygen, a unique behavior that alleviates the hypoxia of TME, making it the most commonly used Mn-based Fenton nanomaterial in biological applications [73]. Moreover, as a classical Fenton-like reagent, MnOx can enhance CDT effectiveness by depleting intracellular GSH. Meanwhile, Mn^2+^ has been demonstrated as an effective contrast agent for tumor *T*_1_-weighted magnetic resonance imaging (*T*_1_-MRI), and it also has excellent PTI, photoacoustic imaging (PAI), and ultrasound imaging capabilities [74,75,76].

In a recent study, Xiao and co-workers reported kinds of macrophage membrane-coated polymer nanogels (MPM@P NGs), which were co-loaded with MnO_2_ nanoparticles and cisplatin to achieve specific targeted chemotherapy and CDT (Figure 5a) [77]. In this nanoplatform, redox-responsive poly (*N*-vinylcaprolactam) nanogels containing disulfide bond (S-S) cross-linkers were disintegrated in TME to specifically release cisplatin and consume GSH to enhance CDT. Furthermore, MnO_2_ simultaneously consumed the highly concentrated endogenous GSH to promote the production of •OH, and the produced Mn^2+^ catalyzed the decomposition of H_2_O_2_ to generate ROS for tumor apoptosis and was also employed for *T*_1_-MRI (Figure 5b). In addition, the outer macrophage membrane enabled the polymer nanogels to penetrate the blood–brain barrier (BBB) due to the overexpressed macrophage-1 antigen, integrin α4, and β1 on the surface for effective glioma targeting. Consequently, this nanoplatform represented a feasible way to enhance MRI-guided synergetic antitumor effect. To realize more accurate cancer imaging and effective treatment, Wang et al. employed poly(lactic-co-glycolic acid) (PLGA) nanoparticles with porous structure as templates to prepare hollow MnO_2_ (HMnO_2_) shells in situ, which were further utilized to deliver bufalin and then cloaked with platelet membrane (Figure 5c) [78]. Bufalin and platelet modification were able to inhibit cancer angiogenesis, block cancer cell growth, and improve tumor targeting mediated by specific binding of P-selectin to the CD44 receptor of cancer cells. Moreover, under conditions of acidic tumor pH and relatively high GSH concentration, HMnO_2_ nanoparticles were rapidly degraded to allow the controlled release of bufalin, while Mn^2+^ was obtained for magnetic resonance monitoring and further targeted chemotherapy and CDT.

Recent reports have especially shown that multicomponent metallic composites improved activity and selectivity compared to single components, which was called the “synergetic catalytic effect” [79,80]. Zhou et al. synthesized cerium peroxide with good stability and then in situ doped Fenton-type metallic peroxides (Fe^2+^, Cu^2+^, Mn^2+^, Ce^4+^, Cr^3+^, etc.) to successfully prepare cerium peroxide matrix (M-CeOx) as a novel Fenton-type bimetallic peroxide nanomaterial [81]. Interestingly, the results of 3,3′,5,5′-tetramethylbenzidine (TMB) assays exhibited that the maximum absorbance increase of TMB increased 40–60-fold at pH 7.0 in the presence of M-CeOx, suggesting that the relatively high pH stimulated the generation of •OH and M-CeOx had excellent catalytic performance. In this prepared M-CeOx, 10% doping of Mn had much higher catalyst efficiency, and Mn-CeOx represented superb anticancer activity at the relatively high pH, due to the in situ self-generated H_2_O_2_ supplement and the multimetal-mediated synergistic-enhanced catalytic activity. In addition, in a recent work by Wu’s group, a novel type of organosilica hybrid micelles (Mn_3_O_4_@PDOMs-GOD) was designed and constructed by co-loading Mn_3_O_4_ and GOD for enhanced ST and CDT [82]. After the entry of Mn_3_O_4_@PDOMs-GOD in the tumor cells, GOD effectively depleted glucose and produced a large amount of H_2_O_2_ at the same time. Under the condition of acidic TME, H_2_O_2_ was decomposed into O_2_ by Mn_3_O_4_, breaking the vicious cycle of hypoxia and simultaneously enhancing the GOD-mediated ST effect through the regeneration of H_2_O_2_. Meanwhile, Mn_3_O_4_ also reacted with overexpressed GSH to release Mn^2+^, thereby converting the increased H_2_O_2_ into highly toxic •OH. Eventually, the Mn_3_O_4_@PDOMs-GOD nanosystem exhibited excellent cascade catalytic performance in the TME, which endowed it with a superior ability of sustained and stable delivery of O_2_ and H_2_O_2_ in situ. Overall, the novel type of organosilica hybrid micelles was a promising approach to improve the efficiency of O_2_-dependent ST and H_2_O_2_-dependent CDT.

## 3. Different Strategies to Improve CDT Performance

Although CDT has unique advantages over cancer treatments, its wide application in vivo is still limited by factors such as harsh Fenton reaction conditions, complex physiological environment, and limited reaction efficiency. Accordingly, to tackle the main obstacles existing in CDT, several strategies have been proposed to change this dilemma in the past few years. In general, the strategies to improve CDT performance are as follows: (1) CDT efficacy is improved by generating more H_2_O_2_; (2) oxidative stress at the tumor cells is enhanced by decreasing the reducing substances amount; and (3) CDT efficiency is improved by reducing the pH value.

### 3.1. Increasing H_2_O_2_ Level

The concentration of H_2_O_2_ plays an important role in the efficiency of CDT. In response to the limitation of endogenous H_2_O_2_ deficiency in the tumor region, an earlier approach was to directly deliver exogenous H_2_O_2_ into the body [83]. For example, Li et al. prepared a PLGA polymersome to encapsulate H_2_O_2_ into a hydrophilic core for direct delivery of exogenous H_2_O_2_ [84]. However, supplementing H_2_O_2_ from an external source had a risk of leakage and was difficult to implement. Therefore, several approaches for production in situ of endogenous H_2_O_2_ have been designed to overcome the limitation of H_2_O_2_ insufficiency in the tumor cells, such as the application of H_2_O_2_-related enzymes, metal peroxides, and photocatalysts.

GOx, which was able to specifically oxidize β-D-glucose into H_2_O_2_ and H^+^, was ideal for increasing tumor acidity and H_2_O_2_ level [85]. Zhang et al. synthesized an adenosine triphosphate (ATP)-responsive autocatalytic Fenton nanoagent (GOx@ZIF@MPN) for tumor therapy (Figure 6a) [86]. The nanoagents had cores which were formed by GOx and then incorporated with ZIF and MPN shells. Interestingly, the MPN shell was decomposed into TA and Fe^3+^ by the overexpression of ATP existing in the tumor cells, resulting in the exposure of internal GOx. Subsequently, the GOx consumed endogenous glucose to generate abundant H_2_O_2_. And TA then reacted with Fe^3+^ to produce Fe^2+^, which converted H_2_O_2_ to •OH and finally exerted good antitumor efficacy. It is widely known that GOx, as an aerobic enzyme, cannot efficiently catalyze the production of H_2_O_2_ in situ due to the hypoxic condition of cancer cells. To address this situation, Feng et al. employed MnO_2_ as an oxygen donor to compose core-shell structured nanoplatform Fe_5_C_2_-GOD@MnO_2_, which generated O_2_ through the weakly acidic TME-specific response, thereby enhancing the process of GOx to catalyze the production of H_2_O_2_ from glucose and O_2_ (Figure 6b) [87]. Tao et al. designed a new CDT nanosystem of RBC@Hb@GOx that effectively overcame the restriction of BBB by using GOx to elevate H_2_O_2_ concentration at the tumor site for the treatment of glioblastoma multiforme. Specifically, hemoglobin (Hb) and GOx were combined as an alternative synthetic Fenton catalyst and then encapsulated with red blood cell (RBC) membranes to achieve longer retention time and less immunogenic responses. 

ROS-regulated nanozymes, such as superoxide dismutase (SOD) and POD, are employed to regulate intracellular and extracellular ROS concentrations. Hence, materials possessing SOD-like or POD-like enzyme activity are extensively utilized to enhance the level of ROS in TME [88,89]. In a study by Sang and her collaborators, the nanoparticle zeolitic imidazole framework-67 (ZIF-67) was constructed by assembling cobalt ions with 2-methylimidazole [90]. Subsequently, small-molecule inhibitor 3-amino-1,2,4-triazole (3-AT) and COOH-PEG-COOH were modified on the surface of ZIF-67 to fabricate the final product PZIF67-AT. PZIF67-AT with SOD-like activity could break the homeostasis of H_2_O_2_ and convert plenty of O_2_^•−^ into H_2_O_2_. At the same time, the small molecule inhibitors released in the acidic microenvironment effectively inhibited the activity of catalase (CAT) and reduced the self-consumption of H_2_O_2_. In a separate study, Dong et al. constructed a nanozyme with tunable multi-enzymatic activities named PHMZCO-AT, which was composed of a hollow mesoporous Mn/Zr-doped CeO_2_ and 3-AT (Figure 6c) [91]. The nanozyme PHMZCO-AT, with enhanced SOD-like and POD-like activities, as well as inhibited CAT-like activity, allowed more H_2_O_2_ to participate in the Fenton reaction, thereby generating massive hydroxyl radicals to induce tumor cells apoptosis and death. 

In addition to the SOD-like activity materials, cisplatin and β-Lapachone have also been demonstrated to be involved in affecting the related enzymes responsible for the generation of H_2_O_2_. For example, cisplatin was an extensively used broad-spectrum chemotherapy drug for cancer therapy. It was reported that nicotinamide adenine dinucleotide phosphate (NADPH) oxidase existing in cancer cells could be specifically activated by cisplatin and, subsequently, an O_2_ molecule was converted into O_2_^•−^ after accepting an electron provided by the oxidation of NADPH, which was further catalyzed by SOD to produce the final H_2_O_2_ [92]. Ren et al. utilized this mechanism to design and synthesize the nanodrug of PTCG, which was composed of phenolic platinum (IV) prodrug, epigallocatechin-3gallate, and copolymer (PEG-b-PPOH) [17]. The study demonstrated that activated cisplatin was employed to increase the intracellular H_2_O_2_ level obviously via a cascade reaction in vitro and in vivo. Moreover, β-Lapachone, a new antitumor drug, was catalyzed by NAD(P)H: quinone oxidoreductase1 (NQO1) to generate H_2_O_2_ [93]. For instance, Wang et al. combined β-Lapachone, iron oxide nanoparticles, and camptothecin (CPT) to form pH- and H_2_O_2_-dual-responsive nanoplatform LaCIONPs (Figure 6d) [94]. The research indicated that β-Lapachone could generate large amounts of H_2_O_2_ through tumor-specific NAD(P)H: NQO1 catalysis. Interestingly, the product H_2_O_2_ not only participated in the Fenton reaction but also specifically activated the release of camptothecin from LaCIONPs for chemotherapy.

**Figure 6 bioengineering-10-00925-f006:**
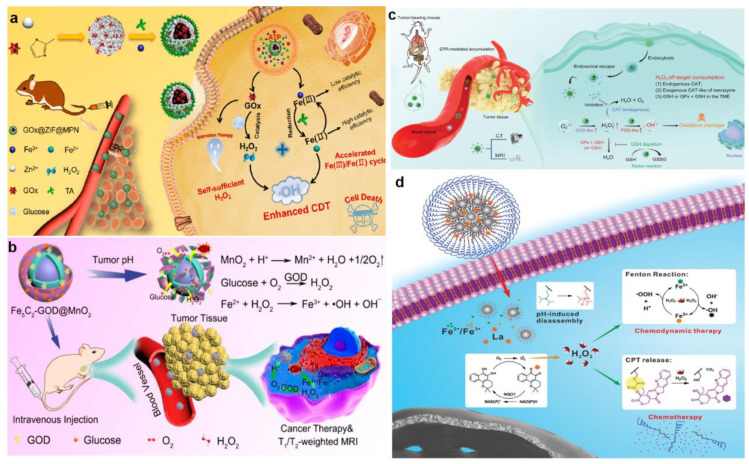
(**a**) Schematic illustration of the synthesis of GOx@ZIF@MPN and its detailed therapeutic mechanisms for synergetic CDT and ST. Reproduced with permission [86]. Copyright 2018, American Chemical Society. (**b**) Schematic illustration of the therapeutic mechanisms of Fe_5_C_2_–GOD@MnO_2_. Reproduced with permission [87]. Copyright 2018, American Chemical Society. (**c**) The action mechanism of PHMZCO–AT nanozymes. Reproduced with permission [91]. Copyright 2021, WILEY–VCH. (**d**) The detailed therapeutic mechanisms of LaCIONPs for chemo/chemodynamic combination therapy. Reproduced with permission [94]. Copyright 2019, WILEY–VCH.

Metal peroxides were conventionally composed of metal ions and peroxo groups. Previous studies have shown that metal peroxides, especially in acidic solutions, more easily react with H_2_O to generate H_2_O_2_, which makes them more suitable for tumor therapy through the acidic TME response [95]. Meanwhile, with the rapid development of nanotheranostic technology in recent years, many nanostructured metal peroxides have been designed and fabricated for biomedical research and have achieved good therapeutic outcomes [49,96,97,98]. As a highly biocompatible metal peroxide, calcium peroxide (CaO_2_) was rapidly hydrolyzed in the acidic TME to produce O_2_ and H_2_O_2_ while releasing large amounts of Ca^2+^. The production of H_2_O_2_ not only strengthened the effect of CDT, but also caused the abnormal function of intracellular calcium channels, which eventually led to calcium overload-mediated cell necrosis. In addition, to prevent the disintegration of CaO_2_ during blood circulation in vivo, Shen et al. utilized pH-responsive ZIF-90 as a protective layer to encapsulate CaO_2_ (Figure 7a) [99]. CaO_2_@ZIF-Fe/Ce6@PEG was co-synthesized by Fe^2+^ and photosensitizer chlorin e6 (Ce6). CaO_2_@ZIF-Fe/Ce6@PEG achieved structural disassembly in the weakly acidic TME, and the exposed CaO_2_ was further hydrolyzed to produce O_2_ and H_2_O_2_, which enhanced Ce6-mediated PDT and Fe^2+^-induced CDT. HA, another material that prevents the breakdown of CaO_2_ during circulation, has been widely used as a tumor-targeting delivery carrier due to its good biocompatibility, biodegradability, and specific binding ability to CD44 receptors [100]. Meanwhile, HA exhibited strong affinity to CaO_2_ and Fe_3_O_4_ based on the coordinating properties of carboxyl groups with Ca^2+^ and Fe^3+^. Therefore, Han et al. prepared CaO_2_-Fe_3_O_4_@HA nanoparticles utilizing HA as a stabilizer and tumor-targeting block to deliver CaO_2_ (Figure 7b) [101]. In addition, there are several other metal peroxides that have shown the ability to generate H_2_O_2_. For instance, ZnO_2_ nanoparticles generated exogenous H_2_O_2_ and Zn^2+^ under the weakly acidic TME, and the generated Zn^2+^ could further promote the generation of endogenous O_2_^•−^ and H_2_O_2_ after entering cancer cells [96,97].

In the fields of energy and the environment, artificial photocatalysis is a common approach to produce H_2_O_2_. In order to apply the approach of photocatalytic synthesis of H_2_O_2_ to tumor treatment, it urgently needed to be considered that the low efficiency of existing photocatalysts to generate H_2_O_2_ under the low oxygen condition, as well as the critical characteristic of the tumor microenvironment (pO_2_ ≤ 5 mmHg), retrained its application. Therefore, Ma et al. proposed a C_5_N_2_ photocatalyst with a conjugated C=N chain by Schiff base reaction for stable and sufficient H_2_O_2_ production in both hypoxic and normoxic conditions (Figure 7c) [102]. The results showed that the CB/VB position of C_5_N_2_ was downshifted remarkably due to the reinforced delocalization of π-electrons, which contributed to a favorable CB and VB band position for H_2_O_2_ photosynthesis in kinetics and thermodynamics (Figure 7d). Encouragingly, the H_2_O_2_ production rate of C_5_N_2_ was 1550 μmol L^−1^ per hour, which was approximately 298 times higher than that of traditional g-C_3_N_4_ (5.2 μmol L^−1^ per hour), indicating that C_5_N_2_ had high photocatalytic H_2_O_2_ production performance (Figure 7e). Since H_2_O_2_ did not have a direct strong killing effect on cancer cells, the original C_5_N_2_ was exfoliated into nanosheets by ultrasonication, and then Fe^3+^ was doped into C_5_N_2_ nanosheets to achieve effective CDT. Overall, this finding proposed a promising approach for improving CDT efficiency.

### 3.2. Decreasing the Level of Reducing Substances

The high expression of GSH in tumor cells hinders the production of ROS, further weakening the antitumor effect of CDT owing to the balance of oxidation and reduction being broken in the presence of high-level GSH in the TME. Therefore, downregulation of GSH in the TME was required to enhance the effectiveness of CDT.

One strategy for eliminating GSH was to deplete the existing GSH in tumor cells. Metals in the oxidized state were the most commonly used oxidants in existing GSH consumption because they could directly interact with GSH and consume it rapidly [103,104]. For instance, Chen et al. designed carrier-free Fe(III)-artemisinin (ART)-coordinated nanoparticles (Fe(III)-ART NPs) via coordination-driven self-assembly for self-enhanced MRI-guided CDT (Figure 8a) [105]. After cellular endocytosis, Fe(III)-ART NPs decomposed in an acidic and redox environment of endo/lysosomes to release Fe^3+^ and ART. The Fe^3+^ was then reduced to Fe^2+^ by intracellular GSH, catalyzing the endoperoxide of ART to generate toxic C-centered free radicals for efficacious CDT. Moreover, significant toxicities were detected in A549 cells co-incubated with Fe(III)-ART NPs. In vitro studies demonstrated that after being treated with 200 μg mL^−1^ Fe(III)-ART NPs for 24 h, the cell viability decreased to 34%, resulting from DNA damage and lipid and protein oxidation (Figure 8b). In addition, FLI was performed using ICG-labeled Fe(III)-ART NPs. The experiment showed that signals at the tumor site were detected within 8 h and maximized at 48 h. Furthermore, T2-weighted MRI signals in tumors were significantly enhanced at 32 h after intravenous administration of Fe(III)-ART NPs, proving that Fe(III)-ART NPs successfully accumulated in tumor cells (Figure 8c). Notably, the free radical generation process of Fe(III)-ART NPs was without reliance on pH or endogenous H_2_O_2_, thereby breaking through the bottlenecks of traditional CDT. Carbon dots (CDs) are small fluorescent carbon nanoparticles less than 10 nm in diameter, and have high chemical stability, luminescence, and broadband optical absorption. Existing studies have shown that CDs easily and rapidly formed iron-containing nanoparticles with Fe^3+^ and then quench the fluorescence. Subsequently, GSH continues to react with the nanoparticles to restore fluorescence while oxidizing into GSSG. In this way, GSH in the tumor region was consumed, and CDs-based GSH FLI was realized for further cancer therapy [106,107]. In recent work, Li et al. combined fluorescence-imaging CD and Fe^3+^ to produce a TME stimuli-responsive therapeutic nanodrug (Fe-CD) for FLI of glutathione depletion and tumor therapy (Figure 8d) [108]. After the entry of Fe-CD into the tumor cells, Fe-CD reacted with GSH and then released Fe^3+^, which not only acted as a quencher but was also an important factor in enhanced CDT. In this process, GSH was converted to GSSG, while Fe^3+^ was simultaneously reduced to Fe^2+^, making CDT more efficient. Moreover, this important process was also monitored by enhancing the fluorescence intensity of Fe-CD. In vitro data showed little change in GSH content when the A549 cells were treated with CD. However, when treated with Fe-CD, GSH content was significantly reduced, by 31.8%, due to the introduction of Fe^3+^, suggesting that Fe-CD had a significant GSH consumption potential (Figure 8e). In addition, this process restored the bright fluorescence of CD, while the reduction of intracellular GSH content was qualitatively indicated by the increase of fluorescence intensity.

The other strategy for more effectively eliminating GSH was to inhibit the synthesis of GSH utilizing chemical inhibitors. However, the use of inhibitors was limited by their high toxicity to normal cells and short half-life. In order to overcome these shortcomings and more effectively eliminate GSH in the tumor, Huang et al. synthesized a multistage GSH exhaustion nanomedicine, named TDMH (TP/2-DG @HMnO_2_@HA), to achieve high-efficiency CDT (Figure 9a) [109]. HA, which was modified on the surface of nanoparticles, enhanced tumor recognition by binding to the CD44 receptor on tumor cells. After HMnO_2_ entered the TME, it underwent a redox reaction with GSH to generate Mn^2+^ and GSSG, thereby quickly depleting the original GSH in the tumor. The generated Mn^2+^ catalyzed the formation of •OH from H_2_O_2_ in the presence of bicarbonate (HCO_3_^−^), which was abundant in the physiological medium. On the other hand, TP and 2-DG were released due to the collapse of the HMnO2 framework. Notably, the released TP targeted Nrf2 (nuclear factor (erythroid-derived 2)-like 2)/SLC7A11, thereby preventing de novo synthesis of GSH. As a commonly used glycolysis inhibitor, 2-DG blocked ATP production in the glycolytic pathway, which provides energy for tumor cells. Consequently, these reaction processes enabled multistage GSH depletion and amplified differences in the susceptibility of tumor cells and normal cells to oxidative stress. 

High endogenous H_2_S expression is unique to the colon cancer tumor microenvironment compared to other tumor types [110]. As such, it is the target of many colon cancer treatments. The reducing capacity of endogenous H_2_S (currently 0.3–3.4 mM) in colon cancer is much stronger than that of GSH [111]. Hence, the efficacy of CDT in colon cancer is limited by endogenous H_2_S, which has a strong reducing ability and can scavenge the generated hydroxyl radicals. To solve the problem, Liu et al. synthesized H_2_S-activated CuFe_2_O_4_ nanoparticles using the characteristic that H_2_S could react with metal oxides and accelerate the valence transformation of Fenton or Fenton-like reagents for efficacious CDT of colon cancer (Figure 9b) [112]. Firstly, CuFe_2_O_4_ nanoparticles reacted with endogenous H_2_S in situ in the TME, leading to the depletion of endogenous H_2_S in colon cancer. Meanwhile, Cu_9_Fe_9_S_16_ nanoparticles with strong NIR absorption were obtained by CuFe_2_O_4_, which were used as photothermal-enhanced CDT agents and intelligent PAI reagents (Figure 9c,d). In addition, parts of Cu^2+^ and Fe^3+^ in CuFe_2_O_4_ nanoparticles were reduced to Cu^+^ and Fe^2+^ by endogenous H_2_S to form hydroxyl radicals via Fenton or Fenton-like reactions. Eventually, CuFe_2_O_4_ had a wide range of promising applications in anticancer drug delivery for colon cancer.
Figure 9(**a**) The therapeutic mechanisms of TDMH nanomedicine. Reproduced with permission [109]. Copyright 2022, American Chemical Society. (**b**) The action mechanism of CuFe_2_O_4_ nanoparticles in cancer cells. (**c**) Thermal imaging of different groups under 808 nm laser irradiation. (**d**) Corresponding PA signals of HCT116 tumor-bearing mice. Reproduced with permission [112]. Copyright 2021, Elsevier Ltd.
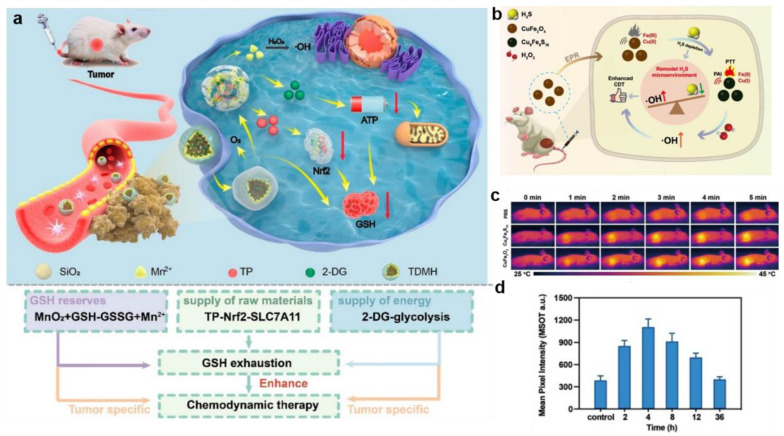


### 3.3. Lowering pH Value

The pH range of the TME was between 6.5 and 7.0, while the optimal pH for Fenton reactions ranged from 2 to 4 [113]. Since the high pH value inhibited the decomposition of nanomaterials and the release of metal ions with catalytic activity, lowering the pH value of TME could effectively improve the catalytic efficiency of the Fenton reaction and enhance the therapeutic efficiency of CDT. In general, introducing exogenous acids or other substances was an efficient and common approach to regulating the pH of the TME. GOx was able to oxidize glucose to H_2_O_2_ and H^+^, which effectively increased the acidity and hydrogen peroxide in the tumor region, thus improving the efficiency of CDT treatment [85]. For instance, Liu et al., for the first time, designed a ternary pillar[6]arene-based nanocatalyst (GOx@T-NPs) employing pillar[6]arene and ferrocene via host–guest interactions to realize the combination of chemotherapy, ST, and CDT (Figure 10a) [114]. After being internalized and endocytosed by cancer cells, the intratumoral glucose was catalyzed by GOx@T-NPs into H^+^ and H_2_O_2_, thereby reducing the pH value inside cancer cells and further achieving high toxicity toward tumor cells. Meanwhile, overexpressed GSH in tumors was consumed by disulfide bonds of GOx@T-NPs, inducing the massive •OH production and the simultaneous release of the antitumor drug CPT. Li et al. reported a cascade catalytic nanoplatform (GOx-NCs/Fe_3_O_4_) for antibacterial CDT toward *Escherichia coli* (Figure 10b) [115]. Specifically, GOx-NCs/Fe_3_O_4_ was constructed by loading GOx onto the surface of covalent-assembled polymer capsules (NCs) through electrostatic interactions, which was successfully achieved to efficiently encapsulate Fe_3_O_4_ and immobilize GOx. Furthermore, GOx was oxidized into H_2_O_2_ in the presence of glucose, increasing in H^+^ concentration. After that, H_2_O_2_ was converted into •OH triggered by Fe_3_O_4_ nanoparticles through the Fenton reaction, thereby destroying the lipoproteins of the bacterial cell wall and causing the death of pathogens. The results of plate counting showed that GOx-NCs/Fe_3_O_4_ represented much higher inhibition efficiency compared with GOx and GOx-NCs, demonstrating its potential for efficient antimicrobial activity (Figure 10c). As a result, this strategy demonstrated that GOx-NCs/Fe_3_O_4_ improved the efficacy of the Fe_3_O_4_-mediated and GOx-introduced CDT for antibacterial applications.

However, the uncontrollable reaction between GOx and glucose during blood circulation could cause damage to normal tissue, so Shi et al. synthesized an acidity-unlocked nanoplatform (FePt@-FeO*x*@TAM-PEG) in order to selectively produce ROS within tumors while remaining silent in normal tissue (Figure 11a) [116]. Since FePt@FeO*x*@TAM-PEG had a hydrophobic shell, it remained silent in blood and normal tissues. After reaching the tumor area, FePt@FeO*x*@TAM-PEG was decomposed and further reacted with H_2_O_2_ to generate •OH due to the pH-responsive transition from hydrophobic to hydrophilic tamoxifen (TAM). Notably, TAM, an anti-estrogen drug, inhibited mitochondrial complex I and further triggered the adenosine monophosphate-activated protein kinase signaling pathway. This process promoted the decomposition of glucose and the accumulation of lactic acid, which increased the acidity of the cancer cells, thus overcoming the limitations of the weak acidity of the tumor. The increased intracellular acidity boosted the release of Fe^2+^ and Fe^3+^ from FePt@FeO*x* nanoparticles, thereby accelerating the H_2_O_2_ degradation and enhancing the antitumor potential of the nanoplatform. Therefore, the problem of insufficient CDT efficiency and potential side effects on normal cells have been effectively solved by the design of the nanodrug.

Carbonic anhydrase IX (CA IX) is a pH-regulating enzyme overexpressed on the cell membrane of most hypoxic tumors and can catalyze the reversible hydration of CO_2_ to generate H_2_CO_3_ and H^+^, participate in tumor extracellular acidosis, and accelerate tumor metastases [117]. CA IX inhibitors (CAI) inhibit extracellular HCO_3_^−^/H^+^ formation, leading to intracellular glycolytic H^+^ accumulation. Inspired by this, Zuo et al. designed and constructed kinds of hollow mesoporous ferric oxide (HMFe) nanocatalysts (MM@HMFe@BS), which were camouflaged by macrophage membrane as well as loaded with CAI to decrease intracellular pH in the TME for self-amplified CDT (Figure 11b) [118]. The camouflage of macrophage cell membrane enabled the nanocatalysts to have immune evasion ability and recognize tumor endothelium cells and cancer cells via α4/VCAM-1 interaction, further promoting tumor chemotactic aggregation. After the efficient internalization into cancer cells, MM@HMFe@BS was specifically degraded to release HMFe and 4-(2-aminoethyl) benzene sulfonamide (BS). Interestingly, the released HMFe not only acted as a highly active atom-exposing Fenton agent to enhance CDT but was also used with high-efficiency MRI to monitor the biodistribution and treatment progress of MM@HMFe@BS. With the specific release of the CAI, CA IX was inhibited by BS, thus inducing intracellular H^+^ accumulation to accelerate the Fenton reaction. Dong and co-workers reported a multifunctional nanosystem of TP_I_@PP_CAI_ to remodel the TME and reinforce the in situ Fenton reaction, which was composed of the inner TP_I_ micelles-loading iron-oxide nanoparticles (IONs) and the outer poly (dopamine-co-protocatechuic acid) (PDA-PA, PP) coating modified with the CAI [119]. Firstly, 4-(2-aminoethyl) benzenesulfonamide, a CA IX inhibitor, inhibited the overexpressed CA IX, thus leading to intracellular acidification. Next, the PP coating was degraded, triggered by the acidification and NIR irradiation, resulting in the disintegration of the inner micellar structure to further release TPGS-PPh3 and IONs. Research showed that TPGS-PPh3, as the endogenous ROS amplifier, was able to target the mitochondria and then interfere with the function, thereby increasing the intracellular ROS basal level for Fenton reactions [120,121,122]. Additionally, based on the good photothermal conversion performance of PDA, the temperature of TP_I_@PP_CAI_ nanoparticles solution increased by more than 15 °C and reached 46.5 °C at a concentration of 0.5 mg mL^−1^ after NIR irradiation for 8 min, indicating that TP_I_@PP_CAI_ had good photothermal performance to enhance Fenton catalytic efficiency and improve the efficiency of CDT.

Jiang et al. synthesized a nanomedicine of FeCNB with high catalytic activity in acidic/neutral conditions, which overcame the limit of firm acidity required for the traditional Fenton reaction (Figure 11c) [123]. The nanomedicine FeCNB was composed of a Fe/Fe_3_C core and mesoporous graphite carbon shell modified by biotin to realize effective CDT by improving the efficiency of the Fenton reaction in acidic/neutral conditions. Specifically, graphitic carbon received electrons from Fe^0^ via a reduction reaction based on the principle of galvanic cells. Meanwhile, Fe^0^ in FeCNB as a reducing agent was oxidized to Fe^2+^, thus consuming H_2_O_2_ to produce •OH in neutral and acidic environments. Moreover, the experimental results confirmed that the photothermal conversion efficiency of FeCNB was 31.4% with 808 nm laser irradiation, revealing that it had an efficient photothermal conversion function to achieve effective PTT. This work offered a new perspective to remodel the pH in the TME for the development of ideal CDT nanomedicine.
Figure 11(**a**) The therapeutic mechanisms of FePt@–FeO*x*@TAM–PEG. Reproduced with permission [116]. Copyright 2021, WILEY–VCH. (**b**) Schematic diagram of the therapeutic mechanisms of MM@HMFe@BS nanomaterials in vivo. Reproduced with permission [118]. Copyright 2022, American Chemical Society. (**c**) Schematic illustrations of the construction and the therapeutic mechanism of FeCNB [123]. Copyright 2022, Elsevier Ltd.
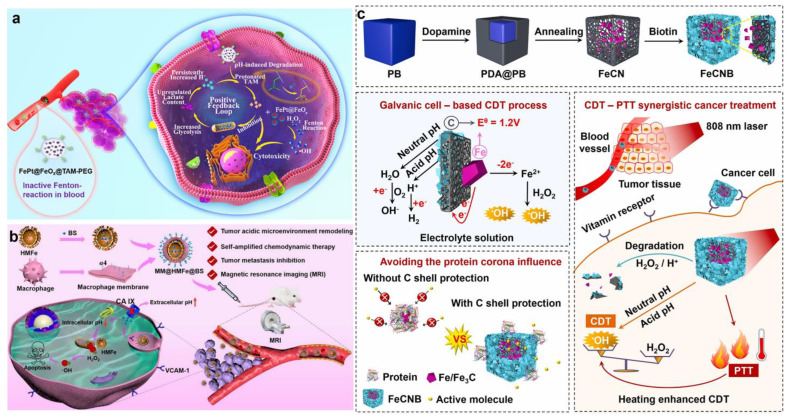


## 4. Conclusions and Future Directions

Cancer has remained a major killer that endangers human life and health. However, traditional cancer treatment techniques such as chemotherapy, radiotherapy, and surgery suffer from poor treatment efficiency, large toxic side effects and easy recurrence, and metastasis, which limit their application in the clinic. Many researchers have focused on exploring more efficient and simpler cancer treatments with important clinical value. Based on Fenton/Fenton-like reactions, CDT is attracting increasing attention as a catalyst-driven modality for cancer therapy because of its remarkable specificity and selectivity for tumors, minimal systemic toxicity and side effects, and the absence of a requirement for field stimulation during treatment. So far, many studies have reported that a large number of metal-based nanocatalytic medicines have been developed to initiate in situ Fenton or Fenton-like reactions in tumor cells, while various nanomaterials have also been used to modulate the tumor microenvironment to improve the therapeutic efficiency of CDT and achieve better tumor treatment effects. Here, Table 1 shows commonly used metal-based catalysts and their applications in biomedicine.

However, CDT is still in its early stages. More research is needed for the clinical translation of CDT, as several important scientific questions need to be addressed. Typical questions are listed below. (1) The target activity of anticancer medicines needs to improve. Metal-based compounds are widely utilized to enhance the performance of CDT based on Fenton and Fenton-like reactions. However, the targeted delivery via the EPR effect is limited, so modifying the surface of nanoparticles for active targeting to recognize tumors selectively could be considered. (2) It is also an urgent issue to reduce the toxic side effects of CDT. Some of these metal ions remain in vivo after therapy, which has potential side effects on the body. Hence, it is necessary to improve the catalytic efficiency of the catalyst and to deepen its ADME (absorption, distribution, metabolism, and excretion) performance study. (3) New smart combinatorial therapeutic nanosystems with high therapeutic efficiency need to be developed. The efficiency of most nanocatalytic therapies depends largely on the tumor microenvironment. The H_2_O_2_ and reductive substances levels, and the pH value in the TME, greatly restrict the efficiency of CDT. Moreover, regulation of the CDT treatment process is also extremely important. Therefore, smart combinatorial therapeutic nanosystems with diagnostic and therapeutic capabilities capable of modulating the tumor microenvironment are a future prospect [28,126]. (4) The efficacy of a single treatment is limited. The further development and design of CDT-induced combination cancer therapies will, to some degree, overcome the shortcomings of CDT, which has a stronger therapeutic performance than any single therapy or their theoretical combinations, such as CDT-PTT, CDT-PDT, CDT-ST, CDT-chemotherapy, CDT-SDT, and CDT-immunotherapy. For example, the Fenton and Fenton-like reactions in CDT are used to produce large amounts of •OH in combination with immunotherapy, and the Russell mechanism in CDT is used to produce a number of ^1^O_2_ to overcome the deficiencies of photodynamic therapy.

CDT is a fascinating research frontier that still requires further development. It is recommended to develop more metal-based and metal-free CDT nanomaterials and to improve the targeting and biological safety of these drugs. At the same time, metal-based nanocatalytic medicines are more likely to progress to the clinical stage, due to their multimodal diagnostic and therapeutic capabilities combined with advanced diagnostic methods like CT imaging and MRI. These nanomedicines could accurately diagnose diseases in real-time while simultaneously providing treatment. Moreover, the ability to monitor the therapeutic effect and adjust the dosage regimen during treatment enables optimal therapeutic outcomes and promotes the application of CDT nanomaterials in clinical transformation [126,127]. Additionally, an in-depth understanding and exploration of the anticancer pathways of CDT at the genetic and molecular levels are needed, which would help deepen the understanding of the anticancer mechanism of CDT and lead to the design of more effective agents.

## Data Availability

Data are contained within the article.

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
