# Peer review of "The Application of Biomedicine in Chemodynamic Therapy: From Material Design to Improved Strategies"

_bioengineering, 2023, doi:10.3390/bioengineering10080925_

Round 1

Reviewer 1 Report

The review paper summarizes the CDT, especially Fenton and Fenton-like reactions, it has reasonable structure and proper figure illustration, I’d support its publication after finishing the following revisions:

1.     The introduction of CDT is inadequate, except for Fenton and Fenton-like reactions to produce ROS, some other mechanisms also work, such as Russell mechanism, which can generate ROS by Cu-MOF (Angew. Chem. 2019, 131, 9951 –9955), please read and include the significant work (maybe set as a single chapter) to make the report complete.

2.     The title looks weird, why not “The Application of Biomedicine in Chemodynamic Therapy:”?

3.     I suggest the authors emphasize the advantages of CDT in the introduction part, especially when comparing it with similar SDT (sound dynamic therapy) or PDT (photodynamic therapy), such as needn’t energy or irradiation from outside.

4.     The direction of words in scheme 1 should be adjusted to make it has better visual effect. And “Mn-based Fenton-like’s reagent” should be changed to “Other metal-based Fenton-like’s reagent”.

5.     In line 53, why not include Fe in the “e.g.” content?

6.     In line 89 and 238, what’s the fate of ·OOH at last? Will it induce cytotoxicity?

7.     Please refer the following important and corresponding works in proper position:

(1)    Nature Nanotechnology volume 2, pages577–583 (2007)

(2)    Tissue Engineering and Regenerative Medicine volume 19, pages237–252 (2022)

Some basic grammar problems such as tense and singular/plural,

line 21: tumor should be tumors;

line 24: discuss should be discusses.

Please check carefully.

Author Response

Reviewer #1:

The review paper summarizes the CDT, especially Fenton and Fenton-like reactions, it has reasonable structure and proper figure illustration, I’d support its publication after finishing the following revisions:

Response: We appreciate the reviewer’s positive comments to our work and detailed suggestions to further improve the technical aspects of this report. We have carefully addressed each question point-by-point and modified the manuscript accordingly.

1.The introduction of CDT is inadequate, except for Fenton and Fenton-like reactions to produce ROS, some other mechanisms also work, such as Russell mechanism, which can generate ROS by Cu-MOF (Angew. Chem. 2019, 131, 9951-9955), please read and include the significant work (maybe set as a single chapter) to make the report complete.

Response: We thank the reviewers for the valuable suggestions, and we reviewed the relevant literature (Angew. Chem. Int. Ed. 2019, 131, 9951-9955; Angew. Chem. Int. Ed. 2019, 58, 18641-18646; ACS Appl. Mater. Interfaces 2021, 13, 50, 59649-59661) in detail and introduced the significant work in the corresponding sections. In fact, according to the Russell mechanism, biohydroperoxides can react with biohydroperoxides to generate 1O2 in the presence of trace metal ions or enzymes, and most biohydroperoxides can be produced in the peroxidation reaction of reactive oxygen species, suggesting their potential to replace photodynamic reagents that depend on light and oxygen.

As suggested by reviewer, and we have adjusted the corresponding discussion and references in the revised manuscript.

Line 50 in revised manuscript:

“Among these treatment modalities, various methods can generate excessive ROS. In PDT treatment, the photosensitizer generates ROS under the action of near-infrared irradiation, and the local ROS burst promotes cell apoptosis. The cavitation effect, sonoluminescence, and piezoelectric effect are the main mechanisms of ROS generation in SDT. At the same time, there are other ways to increase ROS levels in tumors. For instance, Wang et al. reported a kind of ultrathin two-dimensional metal-organic framework (Cu-TCPP), which was endowed with selectively producing 1O2 in the tumor microenvironment for tumor therapy.[10] Firstly, the acidic H2O2 peroxidized the tetrakis(4-carboxyphenyl) porphyrin (TCPP) ligand, and then it was reduced to peroxyl radicals under the action of peroxidase (POD)-like nanosheets and Cu2+. Finally, a spontaneous recombination reaction based on the Russell mechanism occurred to generate 1O2. In addition, Cu-TCPP also consumed glutathione (GSH) through a cyclic oxidation mechanism. Based on the above two "magic weapons", Cu-TCPP nanosheets efficiently and selectively destroyed tumors.

Line 69 in revised manuscript:

“Generally, CDT utilizes peroxidase-like catalysis, metallic catalysis, or Fenton and Fenton-like reactions to significantly enhance intracellular ROS.”

Line 785 in revised manuscript:

“For example, the Fenton and Fenton-like reactions in CDT were used to produce large amounts of •OH in combination with immunotherapy, and the Russell mechanism in CDT was used to produce a number of 1O2 to overcome the deficiencies of photodynamic therapy.”

2.The title looks weird, why not “The Application of Biomedicine in Chemodynamic Therapy:”?

Response: We appreciate the reviewer’s valuable comment and help us to improve this manuscript. According to the suggestions of the reviewers, we have changed the title to “The Application of Biomedicine in Chemodynamic Therapy: From Material Design to Improved Strategies.”

3.I suggest the authors emphasize the advantages of CDT in the introduction part, especially when comparing it with similar SDT (sound dynamic therapy) or PDT (photodynamic therapy), such as needn’t energy or irradiation from outside.

Response: We thank the reviewers for their valuable suggestions and have supplemented the advantages of CDT in the introduction part. It is currently known that CDT can induce cell death by generating reactive oxygen species (ROS) through endogenous chemical energy without the need for external energy inputs, such as through laser irradiation, thus circumventing the limitations of energy forms such as light to penetrate tissues.

Line 12 in revised manuscript:

Chemodynamic therapy (CDT) has garnered significant interest as an innovative approach for cancer treatment, owing to its notable tumor specificity and selectivity, minimal systemic toxicity and side effects, and absence of the requirement for field stimulation during treatment.”

Line 64 in revised manuscript:

“CDT, proposed by Bu and co-workers in 2016, has emerged as a fascinating research area in recent years. This is primarily due to its remarkable specificity and selectivity for tumors, minimal systemic toxicity and side effects, and the absence of a requirement for field stimulation during treatment.[11, 12]”

Line 753 in revised manuscript:

“Based on Fenton/Fenton-like reactions, CDT has attracted increasing attention as a catalyst-driven modality for cancer therapy because of its remarkable specificity and selectivity for tumors, minimal systemic toxicity and side effects, and the absence of a requirement for field stimulation during treatment.”

4.The direction of words in scheme 1 should be adjusted to make it has better visual effect. And “Mn-based Fenton-like’s reagent” should be changed to “Other metal-based Fenton-like’s reagent”.

Response: We appreciate the reviewer’s valuable comment and help us to improve this manuscript. We have adjusted the direction of words in scheme 1 and changed “Mn-based Fenton-like’s reagent” to “Other metal-based Fenton-like’s reagent” in the manuscript.

5.In line 53, why not include Fe in the “e.g.” content?

Response: We sincerely thank the reviewers for their suggestions and fully agree it. As suggested by reviewer, we have added Fe in the “e.g.” content in the corresponding sections. A Fenton reaction is often catalyzed by Fe2+ or Fe3+, which decomposed H2O2 to produce radical •OH. So far, various iron-based nanomaterials have been extensively studied to enhance the efficacy of CDT.

Line 71 in revised manuscript:

“Herein, transitional metal-based nanocatalytic medicines containing some metal ions (e.g., Fe, Cu, Mn, Co, V, Pd, Ag, Mo, Ru, W, Ce) are utilized to release metal ions in tumor cells and then trigger Fenton and Fenton-like reactions to convert H2O2 into •OH with much more toxic in acidic tumor microenvironment (TME), thus inducing the death of tumor cells.[14]”

6.In line 89 and 238, what’s the fate of ·OOH at last? Will it induce cytotoxicity?

Response: Thanks for your suggestion, we have reviewed a lot of literatures and information for our response (Chem. Rev. 2021, 121, 1981-2019; Chem. Eng. J. 2023, 466, 142960). Due to the special electron arrangement structure, O2 are extremely susceptible to the formation of free radicals, and ·OOH is a type of oxygen radical. Oxygen radicals are collectively referred to as ROS along with H2O2, 1O2 and O3. In Fenton and Fenton-like reactions, the produced ·OOH further reacts with high-valent metal ions to produce oxygen. However, throughout the reaction, the rate constant of the reaction of ·OOH and high-valent metal ions (k=0.02 M-1 S-1) is much lower than that of the reaction of ·OOH and low-valent metal ions (k=76 M-1 S-1). According to the suggestions of the reviewers, after consulting the literature and adding the corresponding supplementary discussion in the corresponding chapters.

Line 120 in revised manuscript:

“In this reaction process, the oxidation property of the generated •OOH was lower than that of the generated •OH, and the rate of the generated •OH was relatively faster[27,28].”

Line 269 in revised manuscript:

“As mentioned earlier, the rate of OH production was relatively faster relative to •OOH in Cu-mediated CDT.”

7.Please refer the following important and corresponding works in proper position:

(1) Nature Nanotechnology volume 2, pages577-583 (2007)

(2) Tissue Engineering and Regenerative Medicine volume 19, pages237-252 (2022)

Response: We thank the reviewers for their valuable suggestions and supplement the relevant literatures in proper position. According to the suggestions of the reviewers, after consulting the literature and adding the corresponding supplementary discussion in the corresponding chapters.

Line 474 in revised manuscript:

ROS-regulated nanozymes, such as superoxide dismutase (SOD) and POD, are employed to regulate intracellular and extracellular ROS concentrations. Hence, materials possessing SOD-like or POD-like enzyme activity are extensively utilized to enhance the level of ROS in TME.[88, 89]”

8.Some basic grammar problems such as tense and singular/plural,

line 21: tumor should be tumors;

line 24: discuss should be discusses.

Please check carefully.

Response: We genuinely thank the reviewer’s advice and fully agree with it. According to the suggestions of the reviewers, we have made several revisions to some basic grammatical issues such as tenses and singular/plural to improve this manuscript.

Reviewer 2 Report

The manuscript entitled "The Application of Chemodynamic Therapy in Biomedicine: From Material Design to Improved Strategies" deals with detailed description of a novel cancer treatment strategy. It is well written and organized, however it needs to be revised before it can be considered for publication:

1.      I would like to recommend to include a separate section describing regulatory considerations for chemodynamic therapy

2.      In general, it is well written, but there are some grammars and typing/spacing errors. The English is generally satisfactory but a native speaker should read the paper and correct some sentences

In general, it is well written, but there are some grammars and typing/spacing errorsThe English is generally satisfactory but a native speaker should read the paper and correct some sentences

Author Response

Responses to the comments of Reviewers:Reviewer #2:

The manuscript entitled "The Application of Chemodynamic Therapy in Biomedicine: From Material Design to Improved Strategies" deals with detailed description of a novel cancer treatment strategy. It is well written and organized, however it needs to be revised before it can be considered for publication:

Response: We appreciate the reviewer’s positive comments to our work and detailed suggestions to further improve the technical aspects of this report. We have carefully addressed each question point-by-point and modified the manuscript accordingly.

1.I would like to recommend to include a separate section describing regulatory considerations for chemodynamic therapy

Response: We thank the reviewer for helpful comments, and we reviewed the relevant literature (Chem. Rev. 2021, 121, 1981-2019; Chem. Res. Chinese U. 2022, 38, 481-492; Coord. Chem. Rev. 2021, 429, 213610) and added the discussion about the regulatory considerations for chemodynamic therapy and highlight it according to the suggestions of the reviewers. So far, some multifunctional CDT nanomaterials with imaging modes (such as magnetic resonance, ultrasonic imaging, photothermal imaging, and surface-enhanced Raman) have been developed, which lays a solid foundation to realize the precise theranostics of cancer.

Line 143 in revised manuscript:

“On the other hand, this also contributed to the regulation of the CDT treatment process, minimizing the biotoxicity and improving diagnostic and therapeutic capabilities.”

Line 778 in revised manuscript:

“Moreover, regulation of the CDT treatment process was also extremely important. Therefore, smart combinatorial therapeutic nanosystems with diagnostic and therapeutic capabilities capable of modulating the tumor microenvironment were a future prospect.[28,124]”

Line 791 in revised manuscript:

“At the same time, metal-based nanocatalytic medicines are more likely toprogress to the clinical stage due to their multimodal diagnostic and therapeutic capabilities combined with some advanced diagnostic methods like computed tomography imaging and MRI. These nanomedicines could accurately diagnose diseases in real-time while simultaneously providing treatment. Moreover, the ability to monitor the therapeutic effect and adjust the dosage regimen during treatment enable optimal therapeutic outcomes, and promotes the application of CDT nanomaterials in clinical transformation.[124,125]”

2.In general, it is well written, but there are some grammars and typing/spacing errors. The English is generally satisfactory but a native speaker should read the paper and correct some sentences

Response: We genuinely thank the reviewer’s advice and fully agree with it. According to the suggestions of the reviewers, we have made several revisions to some grammars and typing/spacing issues and corrected some sentences. We believe that the revised manuscript format can meet the strict requirements of magazine as a high-level magazine.

  1. In general, it is well written, but there are some grammars and typing/spacing errors. The English is generally satisfactory but a native speaker should read the paper and correct some sentences

Response: We thank the reviewers for their valuable comments. At the suggestion of the reviewer, we checked the language repeatedly throughout the manuscript. At the same time, we also asked our native English colleagues to help revise the manuscript to meet the requirements of the journal.

Reviewer 3 Report

In this manuscript, “The Application of Chemodynamic Therapy in Biomedicine: From Material Design to Improved Strategies” by Cheng et al. reviews the design and development of different types of nanocatalysts and strategies to improve CDT in recent years, and finally discuss the opportunities and challenges for the future development of CDT. This work is well organized, but with some typos throughout this manuscript. Therefore, I would suggest that authors may take a major revision before publication. Here are the comments and suggestions:

1.        There are many typos throughout this manuscript.

2.        Abbreviations should be defined before their first use, and a list of the abbreviations can be added.

3.        The Table 1 can be revised, and abbreviations can be added as footnotes.

4.        The format of the citation 83 should be corrected.

5.        What would NCs stand for?

6.        All figures can be enlarged.

7.        Figures in Scheme 1 are too small to read.

Author Response

Responses to the comments of Reviewers:Reviewer #3:

In this manuscript, “The Application of Chemodynamic Therapy in Biomedicine: From Material Design to Improved Strategies” by Cheng et al. reviews the design and development of different types of nanocatalysts and strategies to improve CDT in recent years, and finally discuss the opportunities and challenges for the future development of CDT. This work is well organized, but with some typos throughout this manuscript. Therefore, I would suggest that authors may take a major revision before publication. Here are the comments and suggestions:

Response: We appreciate the reviewer’s positive comments to our work and detailed suggestions to further improve the technical aspects of this report. We have carefully addressed each question point-by-point and modified the manuscript accordingly.

1.There are many typos throughout this manuscript.

Response: We thank the reviewers for their valuable comments. We feel very ashamed of the typos in the manuscript. According to the reviewer's suggestion, we have repeatedly checked and corrected typing errors throughout the entire manuscript. At the same time, we also asked our native English colleagues to help revise the manuscript to meet the requirements of the journal.

2.Abbreviations should be defined before their first use, and a list of the abbreviations can be added.

Response: We appreciate the reviewer’s positive comments to our work. According to the suggestions of the reviewers, we have detailed and added a list of the abbreviations in the corresponding sections.

3.The Table 1 can be revised, and abbreviations can be added as footnotes.

Response: We thank the reviewers for the valuable suggestions, and we have revised the Table 1 and supplement the footnotes of it.

Line 98 in revised manuscript:

“PDA: polydopamine; GOx: glucose oxidase; TA: tannic acid; TCPP: (5,10,15,20-tetrakis(4-carboxyphenyl) porphyrin; CPPO: bis(2-carbopentyloxy-3,5,6-trichlorophenyl) oxalate; ZIF-8: zeolitic imidazolate framework-8; DOPA-PIMA-PEG: dopamine and polyethylene glycol decorated poly(isobutylene-alt-maleic anhydride); GOD: glucose oxidase; BSA: bovine serum albumin; R837: immunoadjuvant imiquimod; DOX: doxorubicin; HA: hyaluronic acid; ICG: indocyanine green; PVCL: poly (N-vinylcaprolactam); Ce6: chlorin e6.

4.The format of the citation 83 should be corrected.

Response: We thank the reviewers for the valuable suggestions to our work and fully agree with it. According to the suggestions of the reviewers, we have corrected the format of the citation 83 to improve this manuscript. In addition, we also carefully checked the format of all citations to meet the requirements of the journal.

5.What would NCs stand for?

Response: We appreciate the reviewer’s positive comments to our work and detailed suggestions to further improve the technical aspects of this report. According to the suggestions of the reviewers, we have made appropriate revisions in the corresponding places of the manuscript.

6.All figures can be enlarged.

Response: We thank the reviewer for helpful comments to our work and fully agree with it. According to the suggestions of the reviewers, we have scaled up all figures to the right size in the manuscript.

7.Figures in Scheme 1 are too small to read.

Response: Thanks for your kind advice to our work and we fully agree with it. According to the suggestions of the reviewers, we have resized Figures in Scheme 1 to make them easier to read in the corresponding sections.

Round 2

Reviewer 3 Report

It seems more acceptable now.